# ODYSSEYBENCH: EVALUATING LLM AGENTS ON LONG-HORIZON COMPLEX OFFICE APPLICATION WORKFLOWS

## ABSTRACT

Autonomous agents powered by large language models (LLMs) are increasingly deployed in real-world applications requiring complex, long-horizon workflows. However, existing benchmarks predominantly focus on atomic tasks that are self-contained and independent, failing to capture the long-term contextual dependencies and multi-interaction coordination required in realistic scenarios. To address this gap, we introduce OdysseyBench, a comprehensive benchmark for evaluating LLM agents on long-horizon workflows across diverse office applications including Word, Excel, PDF, Email, and Calendar. Our benchmark comprises two complementary splits: OdysseyBench+ with 300 tasks derived from real-world use cases, and OdysseyBench-Neo with 302 newly synthesized complex tasks. Each task requires agent to identify essential information from long-horizon interaction histories and perform multi-step reasoning across various applications. To enable scalable benchmark creation, we propose HOMERAGENTS, a multi-agent framework that automates the generation of long-horizon workflow benchmarks through systematic environment exploration, task generation, and dialogue synthesis. Our extensive evaluation demonstrates that OdysseyBench effectively challenges state-of-the-art LLM agents, providing more accurate assessment of their capabilities in complex, real-world contexts compared to existing atomic task benchmarks. We believe that OdysseyBench will serve as a valuable resource for advancing the development and evaluation of LLM agents in real-world productivity scenarios.

## 1 INTRODUCTION

Autonomous agents powered by large language models (LLMs) have demonstrated remarkable capabilities across diverse domains, including reasoning (Lin et al., 2024; Boisvert et al., 2024; Yao et al., 2024), software development (Yang et al., 2024; Murty et al., 2024; Zhou et al., 2023; Xie et al., 2025), and scientific research (Drouin et al., 2024; Wu et al., 2025; Zheng et al., 2025). As these agents increasingly transition from research settings to real-world applications, they are expected to handle complex, multi-step tasks such as drafting professional emails, updating documents, and managing personal calendars (Yao et al., 2024; Wang et al., 2024b; Xu et al., 2024a). This shift underscores the need for the development of comprehensive benchmarks that accurately reflect real-world scenarios and rigorously evaluate agent performance in complex, contextual task environments.

However, existing benchmarks for agents predominantly focus on atomic tasks that are self-contained and independent of previous interactions or accumulated context (Zhou et al., 2023; Paranjape et al., 2023; Bonatti et al., 2024; Wang et al., 2024b; Xu et al., 2024a), as illustrated in Figure 1(a). While these benchmarks serve as valuable initial assessments, they fundamentally misrepresent the nature of real-world workflows, which typically unfold across extended periods and encompass various agent-user interactions and require agents to systematically curate, integrate, and leverage information accumulated over extended periods (Schick et al., 2023; Hu et al., 2024; Erdogan et al., 2025). Agents that perform well on atomic task benchmarks may struggle with the contextual dependencies, information persistence, and collaborative workflow management required in real-world scenarios.

In this work, we address these challenges by introducing a novel benchmark **OdysseyBench** designed to evaluate agents on complex, long-horizon workflows spanning diverse office applications, including

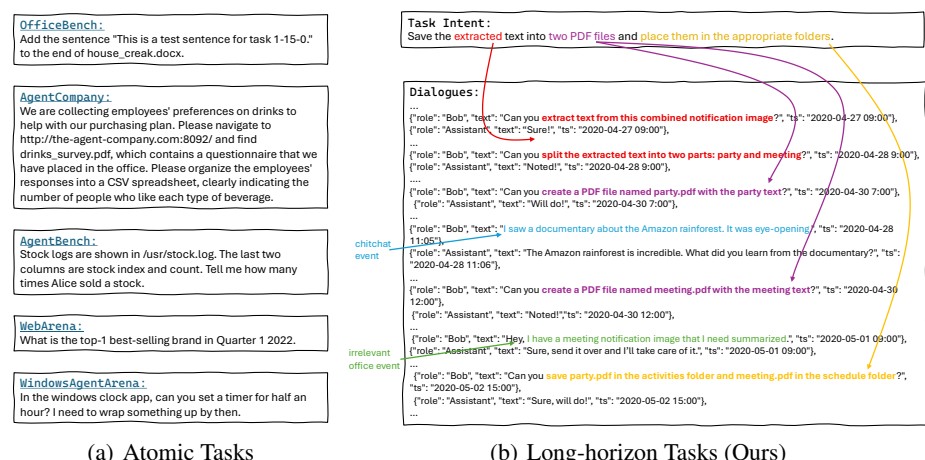

(a) Atomic Tasks        (b) Long-horizon Tasks (Ours)

Figure 1: (a) Atomic tasks: each task is self-contained and does not rely on previous interactions or context. (b) Long-horizon tasks (Ours): a complex task requiring context aggregation, spanning multiple interactions.

■ Word, ■ Excel, ■ PDF, ■ Email, and ■ Calendar. Our benchmark includes two splits: **OdysseyBench+**, which consists of 300 long-horizon tasks originated from real-world use cases in OfficeBench (Wang et al., 2024b), and **OdysseyBench-Neo**, which contains 302 newly generated tasks that are more complex and diverse. Each task, as illustrated in Figure 1(b), is designed to require the agent to reason about the task and extract essential information from long-horizon dialogue histories between the user and agent. This enables the construction of feasible workflows and supports multi-step reasoning across various applications. The tasks are structured to reflect the complexities of agent-user interactions, emphasizing the need for agents to maintain context, synthesize information from prior exchanges, and coordinate actions across diverse tools and environments.

Furthermore, many benchmarks rely on costly human annotation, limiting scalability and constraining the diversity of evaluation scenarios (Zhou et al., 2023; Xu et al., 2024a; Yao et al., 2024). While recent efforts have explored synthetic data generation with LLMs (Ou et al., 2024; Xu et al., 2024b; Xie et al., 2025), these approaches typically yield atomic tasks, lacking the sustained interactions and long-term context essential for realistic workflows. These limitations highlight the urgent need for systematic, automated benchmarks that reflect the challenges of real-world, long-horizon tasks.

To address these challenges, we additionally propose **HOMERAGENTS**, a multi-agent framework that automates the generation of long-horizon workflow benchmarks. Our framework consists of two complementary components: **HOMERAGENTS+**, which leverages existing benchmarks from OfficeBench (Wang et al., 2024b) and employs a two-agent iterative refinement process to transform atomic tasks into contextually rich, multi-interaction scenarios, thereby creating OdysseyBench+; and **HOMERAGENTS-NEO**, which utilizes a multi-agent system operating within realistic application environments to generate entirely new long-horizon tasks from scratch, producing OdysseyBench-Neo. Through systematic environment exploration, task generation, and dialogue creation, HOMERAGENTS enables scalable production of diverse, contextually grounded benchmark tasks that reflect the complexity of real-world scenarios while maintaining the quality standards for rigorous evaluation.

We conduct extensive evaluations of OdysseyBench using state-of-the-art agents. Our experiments reveal that while humans achieve near-perfect performance (over 90% accuracy) on our benchmark, state-of-the-art agents, such as o3 and GPT-5, achieve only around 55% accuracy. This demonstrates that our benchmarks effectively challenge current models and offer a more accurate assessment of their capabilities in real-world contexts.

In summary, our contributions are as follows:

- We introduce **OdysseyBench**, a comprehensive benchmark for evaluating agents on long-horizon workflows across multiple office applications, consisting of **OdysseyBench+** and **OdysseyBench-Neo**.

- We propose **HOMERAGENTS**, a multi-agent framework that automates the generation of long-horizon tasks, enabling scalable and diverse benchmark creation.
- We demonstrate the effectiveness of OdysseyBench in challenging state-of-the-art language agents, providing insights into their performance in complex, real-world scenarios.
- We analyze the impact of dialogue storage formats within OdysseyBench, demonstrating that semantic compression and coherent aggregation are essential for effective multi-step reasoning and agent performance.

## 2 RELATED WORK

**Evaluating LLMs in Executive Environments**   As LLMs advance in tackling real-world challenges (Hurst et al., 2024; Jaech et al., 2024; OpenAI, 2025; Anthropic, 2025b;a; Comanici et al., 2025), there is a growing shift toward evaluating their capabilities in dynamic, executive environments rather than static datasets. Beyond text-based games (Côté et al., 2018; Shridhar et al., 2020), recent research increasingly simulates realistic scenarios to assess agents' proficiency in tool use (Deng et al., 2023; Zhuang et al., 2023; Qin et al., 2023; Lù et al., 2024; Wang et al., 2024a; Shen et al., 2024; Xu et al., 2024a; Sutela & Lindström, 2024). Current benchmarks, such as WebArena (Zhou et al., 2023), AgentBench (Paranjape et al., 2023), WindowsArena (Bonatti et al., 2024), and OfficeBench (Wang et al., 2024b), provide valuable evaluation settings focused on web and office environments. However, these platforms primarily measure atomic performance in self-contained contexts and lack mechanisms to evaluate LLM agents' interactions with complex environments over extended periods. This limitation is significant, as robust assessment of planning, long-term information retrieval, and execution is essential for understanding agents' true capabilities in real-world tasks.

**Synthetic Benchmark Generation**   Existing agent datasets and benchmarks largely rely on human annotators for task creation, demonstrations, and evaluation metric design (Zhou et al., 2023; Xu et al., 2024a; Yao et al., 2024), resulting in high costs and limited diversity. Recent studies try to leverage LLMs to automatically generate agent tasks and trajectories (Ou et al., 2024; Xu et al., 2024b; Xie et al., 2025). For instance, Murty et al. (2024); Pahuja et al. (2025); Trabucco et al. (2025); Gandhi & Neubig (2025) employ LLMs as web agents to synthesize web-based interactions in semi-realistic environments. Moreover, composing atomic tasks is another method to construct more challenging tasks (Boisvert et al., 2024; Drouin et al., 2024). Li et al. (2024) iteratively propose and refine dataset descriptions to generate topic-specific problems. However, these approaches predominantly focus on web-based activities and are generally limited to simple interactions, lacking the complexity of multi-step reasoning and extensive tool use required for robust agent evaluation.

## 3 METHODOLOGY

In this section, we firstly introduce HOMERAGENTS, a multi-agent framework that automatically generates the long-horizon workflow benchmark OdysseyBench in Section 3.1, including two components: HOMERAGENTS+ (Section 3.1.1) and HOMERAGENTS-NEO (Section 3.1.2). We then describe the long-horizon workflow benchmark OdysseyBench in Section 3.2, including the dataset analysis (Section 3.2.2), quality control measures (Section 3.2.3), and human evaluation (**??**).

### 3.1 HOMERAGENTS: AUTOMATING BENCHMARK CREATION

It is highly challenging to create OdysseyBench in a scalable and reliable manner, as it requires generating realistic user–assistant interaction histories and the context-dependent multi-step tasks that reflect the complexity and ambiguity of real-world productivity scenarios. To facilitate this process, we propose a multi-agent framework HOMERAGENTS that automates the generation of tasks, including HOMERAGENTS+ (see Section 3.1.1) and HOMERAGENTS-NEO (see Section 3.1.2).

### 3.1.1 HOMERAGENTS+: STANDING ON THE SHOULDERS OF OfficeBench

HOMERAGENTS+ builds upon the task descriptions from OfficeBench (Wang et al., 2024b) to generate long-horizon dialogue scenarios that more closely mirror real-world productivity workflows. Starting from a given task description $\mathcal{T}$, HOMERAGENTS+ employs a two-agent iterative refinement

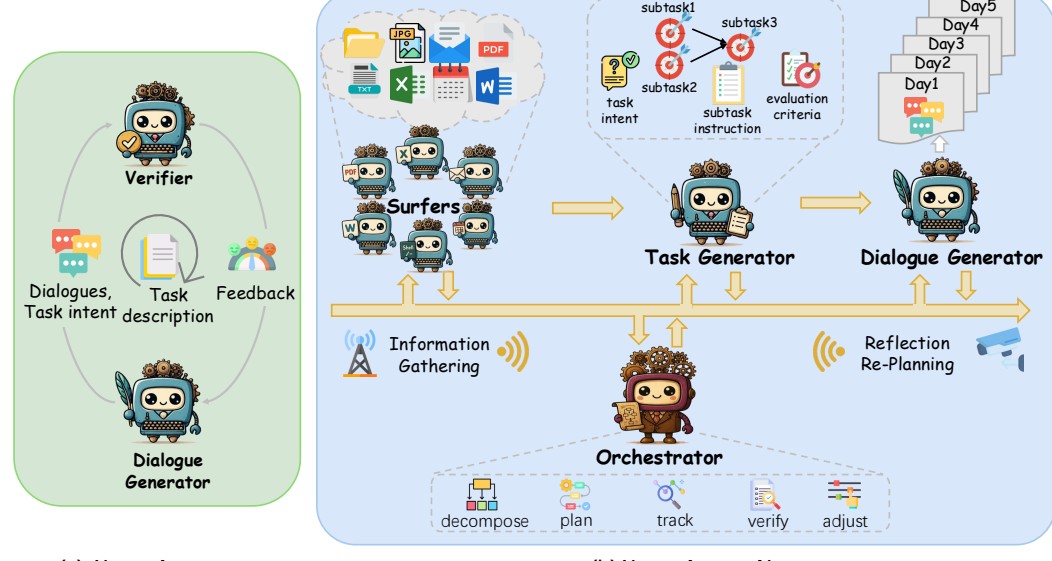

(a) HomerAgents+    (b) HomerAgents-Neo

Figure 2: HOMERAGENTS Framework Overview. HOMERAGENTS consists of two components: HOMERAGENTS+ and HOMERAGENTS-NEO. HOMERAGENTS+ builds upon the task descriptions from OfficeBench to generate long-horizon dialogues, while HOMERAGENTS-NEO creates entirely new tasks and corresponding dialogues from scratch by employing a multi-agent system that operates within realistic application environments.

framework to produce task intents $\mathbb{I}$ and corresponding long-horizon user-assistant dialogues $\mathbb{D}$, thereby contextualizing and enriching the original task.

The framework comprises two core components: a **generator** ($\mathcal{G}$) and a **verifier** ($\mathcal{V}$), as depicted in Figure 2. The generator $\mathcal{G}$ receives the task description $\mathcal{T}$ and any feedback from previous iterations $\mathbb{F}_{i-1}$, and outputs a task intent $\mathbb{I}_i$ along with a corresponding dialogue $\mathbb{D}_i$. Here, the task intent $\mathbb{I}$ succinctly captures the user's goal without specific details, while the dialogue $\mathbb{D}$ provides the natural conversational context leading to the task. The verifier $\mathcal{V}$ then assesses the generated content for dialogue realism, task alignment, and contextual coherence, returning structured feedback $\mathbb{F}_i$.

---

**Algorithm 1:** HOMERAGENTS+

**Input:** Task description $\mathcal{T}$; the generator $\mathcal{G}$; the verifier $\mathcal{V}$; the maximal number of iterations $N_{\max}$;
**Output:** Task intent $\mathbb{I}$ and dialogues $\mathbb{D}$;

1  $\mathbb{F}_0 \leftarrow \varnothing$ ;                                              ▷ Initialize empty feedback
2  **for** $i=1$ to $N_{max}$ **do**
3  $\quad \{\mathbb{I}_i, \mathbb{D}_i\} \leftarrow \mathcal{G}(\mathcal{T}, \mathbb{F}_{i-1})$ ; ▷ The generator $\mathcal{G}$ generates the task intent $\mathbb{I}$ and dialogues $\mathbb{D}$
4  $\quad \mathbb{F}_i \leftarrow \mathcal{V}(\mathcal{T}, \mathbb{I}_i, \mathbb{D}_i)$ ;     ▷ The verifier $\mathcal{V}$ evaluates $\mathbb{I}$ and $\mathbb{D}$, and provides feedback $\mathbb{F}_i$
5  $\quad$ **if** $\mathbb{F}_i ==$ *pass* **then**
6  $\quad\quad$ **return** $\{\mathbb{I}_i, \mathbb{D}_i\}$ ;        ▷ Early stop if the verifier $\mathcal{V}$ thinks the task intent $\mathbb{I}$ and dialogues $\mathbb{D}$ are satisfactory
7  **return** $\{\mathbb{I}_{N_{max}}, \mathbb{D}_{N_{max}}\}$ ;   ▷ Return the task intent $\mathbb{I}$ and dialogues $\mathbb{D}$ after $N_{max}$ iterations

---

This process is executed iteratively, as outlined in Algorithm 1, with a maximum of $N_{\max}$ iterations. In each iteration $i$, the generator $\mathcal{G}$ refines its output based on the original task and accumulated feedback, while the verifier $\mathcal{V}$ either approves the result ("pass") or provides actionable feedback for further improvement. The cycle ends when the verifier approves or the iteration limit is reached. This iterative, feedback-driven approach allows HOMERAGENTS+ to generate realistic, complex long-horizon tasks for rigorous workflow evaluation.

### 3.1.2 HOMERAGENTS-NEO: SCALING UP THE BENCHMARK CREATION

While HOMERAGENTS+ effectively leverages existing benchmarks, HOMERAGENTS-NEO addresses the need for more diverse and scalable task generation by creating entirely new long-horizon tasks from scratch. HOMERAGENTS-NEO employs a multi-agent system that operates within realistic application environments to generate authentic productivity scenarios, as shown in Figure 2.

---

**Algorithm 2: HOMERAGENTS-NEO**

---

**Input:** Applications $\mathcal{A} = \{a_k\}_{k=0}^{K}$; Environment $\mathcal{E}$; Orchestrator $\mathcal{O}$; Surfers $\mathcal{S} = \{S_k\}_{k=0}^{K}$; Task Generator $\mathcal{G}_{\text{task}}$; Dialogue Generator $\mathcal{G}_{\text{dial}}$;

**Output:** Task $\tau$ and dialogue $\mathbb{D}$;

1   **Phase 1: Planning**;
2   $\mathbb{P} \leftarrow \mathcal{O}(\mathcal{A}, \mathcal{E})$ where $\mathbb{P} = \{\mathbb{P}_{\text{surf}}, \mathbb{P}_{\text{task}}, \mathbb{P}_{\text{dial}}\}$;     ▷ Orchestrator drafts the generation plan $\mathbb{P}$
3   **Phase 2: Environment Exploration**;
4   $\mathbb{C} \leftarrow \bigcup_{k=0}^{K} S_k(\mathbb{P}_{\text{surf}}, a_k, \mathcal{E})$;     ▷ Surfers collect contextual information from environment $\mathcal{E}$
5   **Phase 3: Task Generation**;
6   $\tau \leftarrow \mathcal{G}_{\text{task}}(\mathbb{P}_{\text{task}}, \mathbb{C})$ where $\tau = \{\mathbb{T}, \mathbb{I}, \mathbb{K}, \mathbb{E}\}$ ;     ▷ Task Generator generate task components, including task description $\mathbb{T}$, task intent $\mathbb{I}$, subtask instructions $\mathbb{K}$, and evaluation criteria $\mathbb{E}$
7   **Phase 4: Dialogue Generation**;
8   $\mathbb{D} \leftarrow \mathcal{G}_{\text{dial}}(\mathbb{P}_{\text{dial}}, \mathbb{C}, \mathbb{I}, \mathbb{K})$ ;     ▷ Dialogue generator generates T-Days dialogues
9   **return** *Task $\tau$ and dialogues $\mathbb{D}$*;     ▷ Complete task for dataset

---

HOMERAGENTS-NEO consists of **productivity applications** $\mathcal{A} = \{a_k\}_{k=0}^{K}$, **environment** $\mathcal{E}$, **orchestrator** $\mathcal{O}$, **surfers** $\mathcal{S} = \{S_k\}_{k=0}^{K}$, **task generator** $\mathcal{G}_{\text{task}}$, and **dialogue generator** $\mathcal{G}_{\text{dial}}$. Orchestrator $\mathcal{O}$ manages planning, progress tracking, and coordinates the entire generation process by orchestrating each stage of data generation, ensuring coherence in both task and dialogue creation. Surfers $\mathcal{S}$ gather information from environment by interacting with a diverse set of simulated productivity applications. Task generator $\mathcal{G}_{\text{task}}$ synthesizes the tasks and corresponding evaluation criteria. Dialogue generator $\mathcal{G}_{\text{dial}}$ then creates multi-day dialogues simulating realistic user-assistant interactions. The framework consists of four distinct phases, as outlined in Algorithm 2:

**Phase 1: Planning**   The orchestrator $\mathcal{O}$ receives a set of applications $\mathcal{A} = \{a_k\}_{k=0}^{K}$ and environment $\mathcal{E}$, then formulates a generation plan $\mathbb{P} = \{\mathbb{P}_{\text{surf}}, \mathbb{P}_{\text{task}}, \mathbb{P}_{\text{dial}}\}$. This plan specifies how the subsequent phases should explore the environment $\mathbb{P}_{\text{surf}}$, generate tasks $\mathbb{P}_{\text{task}}$, and create dialogues $\mathbb{P}_{\text{dial}}$.

**Phase 2: Environment Exploration**   A collection of specialized surfers $\mathcal{S} = \{S_k\}_{k=0}^{K}$ systematically explore the application environment. Each surfer $S_k$ follows the surfing plan $\mathbb{P}_{\text{surf}}$ to interact with application $a_k$ within environment $\mathcal{E}$, collecting contextual information $\mathbb{C}$. This exploration phase ensures that generated tasks are grounded in realistic application capabilities and user workflows.

**Phase 3: Task Generation**   The task generator $\mathcal{G}_{\text{task}}$ uses contextual information $\mathbb{C}$ and the plan $\mathbb{P}_{\text{task}}$ to produce task specifications $\tau = \{\mathbb{T}, \mathbb{I}, \mathbb{K}, \mathbb{E}\}$, including the task description $\mathbb{T}$, the task intent $\mathbb{I}$, detailed subtask instructions $\mathbb{K}$, and evaluation criteria $\mathbb{E}$. The task description $\mathbb{T}$ outlines the specific goals and requirements of the task, the task intent $\mathbb{I}$ conveys the high-level overall goal but omits specific details of the task, $\mathbb{K} = \{k_1, \ldots, k_t\}$ provides instructions for completing the task, and the evaluation criteria $\mathbb{E}$ define how the task's success will be measured.

**Phase 4: Dialogue Generation**   The dialogue generator $\mathcal{G}_{\text{dial}}$ uses the dialogue plan $\mathbb{P}_{\text{dial}}$, context $\mathbb{C}$, task intent $\mathbb{I}$, and subtask instructions $\mathbb{K}$ to create realistic long-horizon user-assistant conversations $\mathbb{D}$. For each subtask $k_i \in \mathbb{K}$, it generates a dialogue $\mathbb{D}_i$, simulating multi-day interactions, reflecting how the task is approached over multiple days. These are combined into a full dialogue history $\mathbb{D} = \{\mathbb{D}_1, \ldots, \mathbb{D}_t\}$ that illustrates the user's journey through the task, including some task-irrelevant content (e.g., chitchat) to better reflect real-world scenarios. By structuring generation into four phases, HOMERAGENTS-NEO systematically explores application environments and maintains coherence between tasks and dialogues, enabling scalable creation of diverse, realistic benchmark tasks that capture real-world complexity.

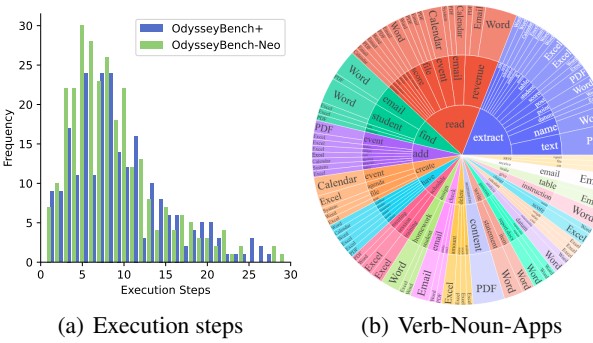

| (a) Execution steps | (b) Verb-Noun-Apps |
|---|---|

Table 1: Human performance of `OdysseyBench-Neo`.

| Task | 1-apps | 2-apps | 3-apps | overall |
|---|---|---|---|---|
| Human | 92.31 | 90.00 | 91.67 | 91.50 |

Figure 3: (1) Execution steps needed for the tasks in `OdysseyBench`. (b) Actions, objects, and applications of `OdysseyBench`.

### 3.1.3 IMPLEMENTATION DETAILS

To balance performance and cost, all agents in HOMERAGENTS use the GPT-4.1 model for strong reasoning at reasonable expense. We set the maximum iterations $N_{max}$ in Algorithm 1 to 5, and generate at least $T = 5$ days of dialogues in Algorithm 2 to capture long-term workflow complexity. HOMERAGENTS-NEO is implemented on the Magentic-One framework (Fourney et al., 2024).

During dialogue generation, the assistant simulates task execution by generating responses based on task descriptions and context, rather than performing real actions. This enables scalable, diverse, and realistic dialogue generation. By deferring execution, our benchmark focuses on agents' ability to curate and integrate information across multiple dialogue turns and days, which is crucial for evaluating long-horizon comprehension and planning.

### 3.2 `OdysseyBench`: LONG-HORIZON WORKFLOW BENCHMARK

#### 3.2.1 EVALUATION

We build `OdysseyBench` in a Docker environment with pre-installed applications, automate operations using Python, and manage documents, emails, and calendar events via a file system. After the agents complete each task, we save the file system and perform customized evaluations to verify correctness.

Our evaluation combines exact matching, fuzzy matching, and execution-based methods. Exact and fuzzy matching check if the agent's output matches the expected results (e.g., keyword matching for documents and calendar events), while execution-based evaluation uses code to verify outputs (e.g., checking calendar conflicts). A task is successful if all criteria are met. We report the **pass rate** as the percentage of tasks completed successfully: $\frac{\#\text{successful tasks}}{\#\text{total tasks}}$.

#### 3.2.2 DATASET ANALYSIS

We provide the statistical analysis of our dataset in Appendix A. We further analyze the distribution of execution steps in `OdysseyBench` (Figure 3(a)), finding that most tasks in both datasets require 3-15 execution turns. This indicates that `OdysseyBench` tasks are sufficiently complex and reflect real-world multi-step workflows. We also examine task diversity in `OdysseyBench`, summarizing actions, objects, and applications in Figure 3(b). The benchmark covers a broad range of actions, objects, and applications, ensuring it captures the complexity and variety of real-world productivity tasks and making it a valuable resource for evaluating long-horizon workflow understanding in LLMs.

#### 3.2.3 QUALITY CONTROL

**Automated Filtering** After creating the initial benchmark using HOMERAGENTS, we implement a multi-step LLM-based filtering mechanism to ensure the quality and reliability of the generated tasks:

1. **Task Evaluation Check**: We firstly verify that each generated task is associated with a well-defined evaluation criteria $\mathbb{E}$. If the evaluation criteria are not supported by our system as described in Section 3.2.1, the task is discarded. This step ensures all the tasks in the benchmark can be objectively assessed for correctness and completeness.
2. **Task Solvability Check**: We prompt a state-of-the-art LLM (e.g., o3) to attempt solving each generated task using using either the full description $\mathbb{T}$ or just the intent $\mathbb{I}$ and subtask instructions $\mathbb{K}$. Ideally, the agent should be able to complete the task if the full description is provided. If the agent fails to complete the task even with the full description, the task is deemed unsolvable and is removed from the benchmark. This step helps eliminate tasks that are inherently flawed or too ambiguous for practical completion.
3. **General Quality Check**: After the previous two checks, we ensure that the tasks in the benchmark are both verifiable and solvable. We then conduct a final quality check using a group of five LLM agents. Each agent independently assess the remaining tasks based on the quality verification guidelines outlined in Appendix B. If a task receives negative feedback from three or more agents, it is removed from the benchmark. This collective evaluation helps maintain high standards for task quality and relevance.

**Human Verification and Post-Editing**   We also implement human verification and post-editing to further enhance the quality of the generated task intent and dialogues. A team of three native English-speaking annotators manually reviews the generated task intent and dialogues, assessing them based on the quality verification guidelines outlined in Appendix B. Due to the complexity of the guidelines, we organize a training session to ensure the annotators fully understand the criteria. During this process, each example in the benchmark is evaluated by all three annotators and further revised if any disagreements arise. The inter-annotator agreement is measured using Fleiss' Kappa score (Fleiss & Cohen, 1973), which is 0.72, indicating substantial agreement among the annotators.

### 3.2.4 HUMAN PERFORMANCE

To establish an understanding of human performance on OdysseyBench, we employ two experienced productivity application users to complete a randomly selected subset of 100 tasks from OdysseyBench-Neo. Each user is instructed to complete the tasks with the full dialogue history $\mathbb{D}$ and task intent $\mathbb{I}$. They are given up to 10 minutes to complete each task and allowed to use any external tools, such as AI writing assistants, to aid in task completion. As shown in Table 1, the human users achieve an overall pass rate of 91.50%, demonstrating that the tasks in OdysseyBench are solvable by humans and providing a benchmark for evaluating LLM performance.

## 4   EXPERIMENTAL SETUP

**Long-Context Evaluation:** We evaluate agent performance on OdysseyBench by providing the entire dialogue history. **RAG Evaluation:** We also assess agents in the Retrieval-Augmented Generation (RAG) setting, where relevant context is retrieved from dialogue history using embedding models. We test two types of stored context: (1) **raw** and (2) **summarized**, each at two granularities. For raw context: (a) *session-level* (entire session as one document), (b) *utterance-level* (each turn as a separate document). For summarized context: (a) *session-level* (session summarized as one document), (b) *chunk-level* (multiple sessions segmented and summarized in chunks).

**Metrics and Models**   As in Section 3.2.1, we use **pass rate** (percentage of successful task completions) as the main metric. We evaluate proprietary LLMs (o3, o3-mini, GPT-4o, GPT-4o-mini, GPT-4.1, GPT-5, GPT-5-chat) and open-weight LLMs (DeepSeek-R1, DeepSeek-R1-Distill-Qwen-32b, Qwen3-32b). The RAG embedding model is OpenAI text-embedding-3-large.

## 5   EXPERIMENTAL RESULTS

**Tasks get increasingly complex with more applications involved, leading to a performance drop.**   As shown in Table 2, performance consistently declines as the number of applications per task increases. For OdysseyBench+, the average performance drops from single-app scenarios to three-app scenarios across all models: o3 drops from 72.83 to 30.36, GPT-4.1 from 55.91 to

Table 2: Performance of the long-context configuration on `OdysseyBench+` and `OdysseyBench-Neo` tasks. We divide the tasks into "1/2/3-apps", specifying the number of applications required by the tasks. The overall performance is reported as the macro-average across all tasks.

| | OdysseyBench+ | | | | OdysseyBench-Neo | | | |
|---|---|---|---|---|---|---|---|---|
| | 1-apps | 2-apps | 3-apps | **overall** | 1-apps | 2-apps | 3-apps | **overall** |
| **Proprietary Models** | | | | | | | | |
| o3 | 72.83 | **70.53** | **30.36** | **56.19** | 68.33 | 60.56 | **59.06** | **61.26** |
| o3-mini | 38.04 | 20.00 | 15.18 | 23.75 | 71.67 | 39.44 | 45.61 | 49.34 |
| GPT-4o-mini | 30.11 | 22.11 | 7.14 | 19.00 | 65.00 | 33.80 | 29.83 | 37.75 |
| GPT-4o | 47.31 | 42.11 | 15.18 | 33.67 | **75.00** | 47.89 | 45.61 | 51.99 |
| GPT-4.1 | 55.91 | 43.16 | 12.50 | 35.67 | **75.00** | 63.38 | 47.37 | 56.62 |
| GPT-5-chat | 55.91 | 48.42 | 20.54 | 40.33 | **75.00** | 57.75 | 51.46 | 57.62 |
| GPT-5 | **75.27** | 66.32 | 25.89 | 54.00 | 61.67 | 56.34 | 53.80 | 55.96 |
| **Open-weight Models** | | | | | | | | |
| DeepSeek-R1 | **53.76** | **47.37** | **20.54** | **39.33** | 78.33 | 60.56 | 44.44 | 54.97 |
| DS.-Distill-Qwen-32b | 30.11 | 16.84 | 1.79 | 15.33 | 40.00 | 22.54 | 10.53 | 19.21 |
| Qwen-3-32b | 38.71 | 33.68 | 11.61 | 27.00 | 41.67 | 22.54 | 21.05 | 25.50 |

Table 3: Performance of RAG-based GPT-4o on `OdysseyBench`. "Long-context prompting" refers to evaluation in the long-context setting. "top-k" denotes the number of top retrieved documents used as context, and "tokens" indicates the total tokens in the retrieved documents.

| storage | granularity | top-k | OdysseyBench+ | | | | | OdysseyBench-Neo | | | | |
|---|---|---|---|---|---|---|---|---|---|---|---|---|
| | | | tokens | 1-apps | 2-apps | 3-apps | overall | tokens | 1-apps | 2-apps | 3-apps | overall |
| Long-context prompting | | | 8000 | 47.31 | **42.11** | 15.18 | **33.67** | 6700 | **75.00** | 47.89 | 45.61 | 51.99 |
| raw | session | 5 | 750 | 40.86 | 40.00 | 11.61 | 29.67 | - | - | - | - | - |
| | | 10 | 1500 | 39.79 | 40.00 | 14.29 | 30.33 | - | - | - | - | - |
| | utterance | 5 | 80 | 29.03 | 35.79 | 8.04 | 23.33 | 90 | 30.00 | 16.90 | 8.19 | 14.57 |
| | | 10 | 155 | 27.96 | 33.68 | 8.93 | 22.67 | 180 | 31.67 | 16.90 | 11.11 | 16.56 |
| | | 25 | 370 | 39.79 | 35.79 | 12.50 | 28.33 | 450 | 35.00 | 32.39 | 21.05 | 26.49 |
| | | 50 | 730 | **57.69** | 40.00 | 17.17 | 29.41 | 915 | 56.67 | 40.85 | 31.58 | 38.74 |
| summarized | session | 5 | 290 | 29.03 | 35.79 | 9.82 | 24.00 | 2200 | **75.00** | 46.48 | 49.12 | 53.64 |
| | | 10 | 650 | 33.33 | 36.84 | 9.82 | 25.67 | - | - | - | - | - |
| | chunk | 5 | 290 | 30.11 | 29.47 | 12.50 | 23.33 | 1200 | 30.11 | 29.47 | 12.50 | 23.33 |
| | | 10 | 380 | 40.86 | 34.74 | 16.96 | 30.00 | 1260 | 40.86 | 34.74 | 16.96 | 30.00 |
| | | 25 | 600 | 46.24 | 36.84 | **19.64** | 33.33 | 1360 | 68.33 | 59.16 | 50.88 | **56.29** |
| | | 50 | 670 | 44.09 | 40.00 | 16.96 | 32.67 | 1460 | 68.33 | 59.16 | 48.54 | 54.97 |

12.50, and DeepSeek-R1 from 53.76 to 20.54. A similar but less pronounced trend appears in `OdysseyBench-Neo`. For instance, o3 maintains relatively stable performance (68.33 to 59.06), while GPT-4o shows a decline from 75.00 to 45.61. This highlights the challenge LLMs face in coordinating information across applications, which requires advanced reasoning about dependencies and state.

**More context typically leads to better performance, but at a cost.** As shown in Table 3, storing raw data without retrieval (long-context prompting) gives the highest performance (33.67 on `OdysseyBench+` with 8000 tokens; 51.99 on `OdysseyBench-Neo` with 6700 tokens), but uses many tokens. Utterance-level retrieval in RAG offers a good balance, peaking at 29.41 with 730 tokens on `OdysseyBench+` and 38.74 with 915 tokens on `OdysseyBench-Neo`. It outperforms the long-context prompting for some `OdysseyBench+` tasks but underperforms in `OdysseyBench-Neo`, likely due to shorter dialogues in `OdysseyBench+` and excessive fragmentation in `OdysseyBench-Neo` (see dataset statistical analysis in Appendix A). This highlights the need to maintain coherent conversational boundaries, as fragmented utterances can undermine context integrity.

**Summary storage effectively captures task essence.** Summarization improves performance by condensing information and retaining key context. Session-level summaries outperform the long-context prompting (53.64 on `OdysseyBench-Neo` with one third of the tokens), while chunk-level

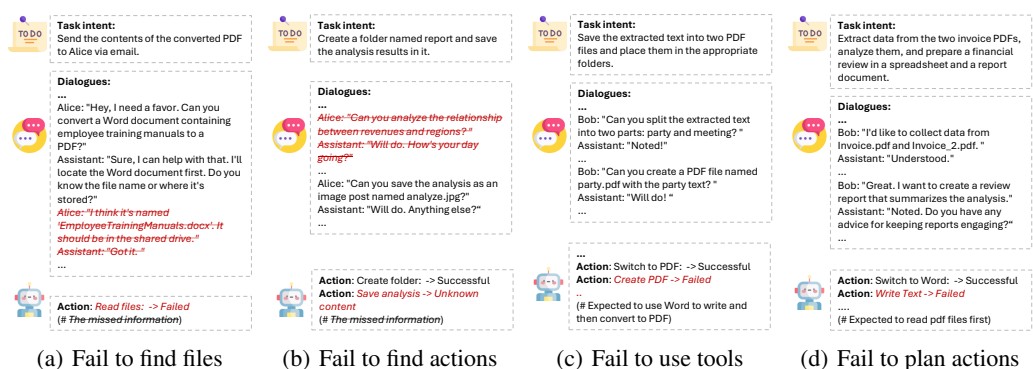

| (a) Fail to find files | (b) Fail to find actions | (c) Fail to use tools | (d) Fail to plan actions |

Figure 4: Typical failure cases of the LLM agents in `OdysseyBench`.

summaries do even better (56.29 with less than 20% of the tokens). Summarized context distills essential information, removes redundancy, and increases semantic density, enabling more efficient and precise retrieval within the same token budget. However, Increasing top-k from 25 to 50 slightly reduces performance (56.29 to 54.97 on `OdysseyBench-Neo`), indicating that more context can add noise and irrelevant information. Quality of retrieved content matters more than quantity. These results highlight the need for memory architectures that emphasize semantic aggregation and context continuity for complex, multi-step workflows.

## 6 CASE STUDY

To analyze LLM agent failures in `OdysseyBench`, we manually reviewed execution traces and categorized errors into four main types: (1) **Missing required files**: Agents overlook input files mentioned in the dialogue (e.g., missing "EmployeeTraining-Manuals.docx" in Figure 4(a)). (2) **Missing required actions**: Agents fail to perform or modify files as instructed (e.g., omitting the "analyze the relationship" step in Figure 4(b)). (3) **Incorrect tool calls**: Agents use the wrong tool or arguments (e.g., creating PDFs directly instead of converting from Word in Figure 4(c)). (4) **Inaccurate planning**: Agents lack a coherent plan, such as writing in a Word document before reading the necessary PDF content (Figure 4(d)). Further quantitative analysis based on the file types involved in failed executions (Figure 5) reveals that most errors are associated with file creation or writing tasks, particularly for formats such as "docx" and "xlsx". This indicates agents often struggle with complex, multi-step workflows that require precise coordination of tools, timing, and reasoning.

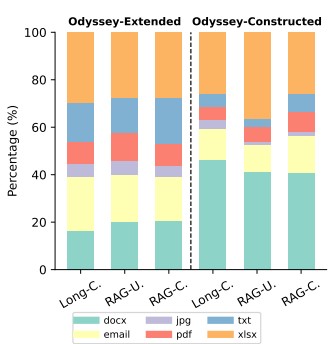

Figure 5: Errors on file types with three configurations: long-C. (long-context), RAG-U. (RAG-utterance), and RAG-C. (RAG-chunk).

## 7 CONCLUSION

In this work, we addressed the critical limitation of existing atomic task benchmarks by introducing `OdysseyBench`, a comprehensive benchmark for evaluating language agents on long-horizon workflows across diverse office applications. Our key contribution, HOMERAGENTS, provides a scalable multi-agent framework that automates benchmark generation through two complementary approaches: HOMERAGENTS+ transforms existing atomic tasks into contextually rich scenarios to create `OdysseyBench+`, while HOMERAGENTS-NEO generates entirely new complex tasks from scratch to produce `OdysseyBench-Neo`. Extensive evaluation revealed substantial performance gaps between state-of-the-art agents on our benchmark compared to atomic tasks, demonstrating the importance of contextual dependencies and multi-interaction coordination in realistic scenarios.

## ETHICS STATEMENT

This work introduces `OdysseyBench` and the HOMERAGENTS multi-agent framework for benchmarking LLM agents on long-horizon office application workflows. All experiments were conducted using publicly available models and datasets, or proprietary models accessed under their respective licenses and terms of use. No human subjects or private user data were involved in this research. The benchmark tasks and dialogues were generated synthetically or derived from existing public datasets, with explicit guidelines to avoid personal, sensitive, or inappropriate content. We encourage responsible use of `OdysseyBench` and HOMERAGENTS, with attention to fairness, transparency, and the limitations of underlying models and synthetic data.

## REPRODUCIBILITY STATEMENT

We are committed to reproducibility in this work. Detailed descriptions of the `OdysseyBench` benchmark, task generation algorithms, and evaluation protocols are provided in Section 3 and throughout the main paper. Experimental setups, including model configurations, dataset splits, evaluation metrics, and implementation details, are thoroughly documented in Section 4 and Appendix. All datasets used are either publicly available or will be released with the benchmark. To further support reproducibility, we will release the `OdysseyBench` benchmark, HOMERAGENTS framework, and code for all experiments upon publication, enabling other researchers to replicate our results and build upon this work.

## THE USE OF LARGE LANGUAGE MODELS (LLMS)

In preparing this work, we utilize large language models (LLMs) as general-purpose tools to assist with writing polish and grammar correction. The LLMs are not involved in research ideation, experimental design, or substantive content generation. Their role is limited to improving the clarity and readability of the text, ensuring grammatical accuracy, and refining the presentation of our findings. All scientific contributions, analyses, and conclusions are solely the work of the authors.

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

Table 4: Data statistics of `OdysseyBench+` and `OdysseyBench-Neo`.

| | OdysseyBench+ | | | | OdysseyBench-Neo | | | |
|---|---|---|---|---|---|---|---|---|
| | single apps | two apps | three apps | **overall** | single apps | two apps | three apps | **overall** |
| Total # conversation $h$. | 93 | 95 | 112 | 300 | 60 | 71 | 171 | 302 |
| Avg. # session $k$. in conversation $h$ | 27.8 | 24.7 | 30.6 | 27.9 | 5.0 | 5.0 | 5.1 | 5.0 |
| Avg. # utterance $j$. in session $k$ | 10.8 | 12.1 | 11.4 | 11.4 | 72.3 | 73.5 | 73.3 | 73.2 |
| Avg. # tokens. conversation $h$ | 3323.2 | 3209.6 | 3809.9 | 3468.9 | 5031.6 | 5223.1 | 5196.4 | 5169.9 |
| Avg. # tokens. sessions $k$ | 119.7 | 130.1 | 124.4 | 124.6 | 1006.3 | 1041.7 | 1026.1 | 1025.8 |
| Avg. # tokens. utterance $j$ | 11.1 | 10.8 | 10.9 | 10.9 | 13.9 | 14.2 | 14.0 | 14.0 |

## A  DATASET STATISTICAL ANALYSIS

As shown in Table 4, our dataset comprises 602 tasks, categorized by the number of applications involved: Single App (153 tasks), Two Apps (166 tasks), and Three Apps (283 tasks). Each task is documented through multi-day dialogues, with at least five days per task. Dialogues occurring within the same day are grouped into a single session, and every dialogue contains a minimum of 10 utterances, ensuring rich interaction data. `OdysseyBench+` contains 300 conversation histories with an average of 27.9 sessions per conversation and 11.4 utterances per session, resulting in relatively short sessions with an average of 124.6 tokens per session. In contrast, `OdysseyBench-Neo` comprises 302 conversations with a more structured format of exactly 5 sessions per conversation (corresponding to the 5-day dialogue design) but significantly longer sessions, averaging 1025.8 tokens each and 73.2 utterances per session. This design difference reflects `OdysseyBench-Neo`'s focus on creating more comprehensive daily interactions, while `OdysseyBench+` maintains the original fragmented conversation structure from `OfficeBench`. Overall, `OdysseyBench-Neo` generates richer conversational content with approximately 49% more tokens per conversation (5169.9 vs. 3468.9 tokens), demonstrating the enhanced depth and complexity of the newly generated tasks.

## B  QUALITY VERIFICATION GUIDELINE

To ensure consistency and quality, we design a quality verification guideline following the best practices in the AI community. Annotators are instructed to remove any task or dialogue that does not meet **all** of the following criteria:

- **Completeness**: The combination of task intent and dialogue must provide all information necessary for a competent agent (or human) to complete the task. No essential details should be missing from the dialogue history or task intent.
- **Soundness (No Information Leakage)**: The task intent must not reveal specific details from the original task description that are intended to be discovered through the dialogue. All critical information for task completion should be conveyed through the dialogue, not leaked in the intent.
- **Clarity and Coherence**: The task description, intent, and dialogues must be clearly written, logically structured, and free of ambiguity. Dialogue turns should follow a natural, realistic conversational flow, with each utterance making sense in context.
- **Solvability**: The task must be solvable using only the information provided in the intent and dialogue, without requiring external knowledge or assumptions. There should be no contradictions or missing steps that would prevent successful completion.
- **Relevance and Appropriateness**: The task and dialogue should be relevant to real-world productivity scenarios and appropriate for the intended application environment. Content must be free from offensive, biased, or inappropriate language.
- **Diversity and Realism**: Dialogues should include a mix of task-relevant and occasional task-irrelevant (e.g., chitchat) content to reflect real-world interactions, but should not be dominated by irrelevant content. Tasks should not be trivial or repetitive; they should reflect the complexity and variety expected in real-world workflows.
- **Language Quality**: All text must be grammatically correct, fluent, and written in natural English.

Table 5: The number of execution steps of the task in `OdysseyBench+` and `OdysseyBench-Neo` under different configurations indicates how many steps are required to successfully execute the task. "configuration" represents the experimental setup used for evaluation.

|  | configuration | 1-apps | 2-apps | 3-apps | overall |
|---|---|---|---|---|---|
| OdysseyBench+ | long-context | 6.31 | 11.61 | 12.70 | 10.25 |
|  | RAG-utterance | 6.85 | 11.48 | 14.70 | 11.05 |
|  | RAG-chunk | 7.25 | 8.28 | 14.86 | 10.10 |
| OdysseyBench-Neo | long-context | 7.81 | 9.63 | 11.74 | 10.46 |
|  | RAG-utterance | 8.17 | 9.66 | 12.52 | 10.92 |
|  | RAG-chunk | 7.93 | 9.92 | 12.54 | 10.95 |

Tasks or dialogues failing to meet any of these standards are removed. This process ensures the benchmark remains high-quality, challenging, and representative of real-world use cases, in line with accepted practices in the AI community.

## C  ANALYSIS OF EXECUTION STEPS

Furthermore, analysis of execution steps in Table 5 reveals that chunk-level summaries introduce negligible computational overhead, and in some cases, even reduce the number of steps required to complete tasks. This indicates that summarization not only boosts performance, but also streamlines the reasoning process by providing relevant context efficiently, without overwhelming the model. These findings underscore the critical role of semantic compression and coherent aggregation in enabling effective multi-step reasoning.

## D  CRITERIA OF VERIFIER AGENT

We provide the criteria used by the verifier agent in HOMERAGENTS+ to ensure the quality and realism of the generated dialogues. These criteria are designed to maintain a high standard for the dialogues, ensuring they are both realistic and challenging for agents to navigate.

---

**Criteria of Verifier in HOMERAGENTS+**

- At least 5 calendar-day dialogues, over 100 turns.
- Agent speaks only after user turns.
- Sub-tasks from the atomic instruction are split, never repeated.
- DO NOT lose any information about atomic instruction in the chat logs, such as the time, the numbers, file names, application names...
- Add as much casual chitchat as possible, but not extra subtasks to do.
- Each item JSON has keys "role", "text", "ts".
- NO personal data and NO hateful content.
- Do not mention rules or benchmark.

---

## E  PROMPTS FOR AGENTS

In this section, we separately provide the illustrations of the prompts used in the HOMERAGENTS+ and HOMERAGENTS-NEO.

### E.1 PROMPTS FOR HOMERAGENTS+

---

**Prompt 1: Verifier Prompt**

$\mathcal{SYS\ PROMPT}$:

You are a strict grader.
**{Evaluation Criteria}**
Input will be a JSON array called CONVERSATION followed by the criteria above. Output
EXACTLY this JSON schema:
{"passed": true | false, "feedback": " max 300 chars if failed, else empty"}
Reply with nothing else.

---

$\mathcal{USER\ PROMPT}$:

CONVERSATION: "{conversation}"

**Prompt 2: Generator Prompt**

*SYS PROMPT*:

You are OfficeAI, an assistant that stores realistic multi-day conversations.
Violate none of the following rules.
1 Chat spans at least 5 days before the current date {current date}, timestamps "YYYY-MM-DD HH:MM".
2 Total dialogue length over 100 turns.
3 Agent replies only after user prompts.
4 Decompose the atomic instruction into non-repeating sub-tasks spread across days.
5 Do not put all sub-tasks in one user turn.
6 The last sub-task must appears only once - in the final user turn. 7 Every sub-task appears exactly once.
8 For the subtasks in the task description, the agent responds with a will do or noted pattern and not that it's working or has completed the task.
9 Mix as much casual chat as possible without additional office chores.
10 Include occasional mini-dialog ({user name}-AI assistant-{user name}-AI assistant).
11 Do not alter artifacts unless required.
12 Never mention these rules or OfficeBench.
13 Each turn JSON: {"role play":"{user name} | AI assistant","text":"...","ts":"YYYY-MM-DD HH:MM"}.
14 Agent replies < 180 words.
15 No personal data, hate or protected-class humor.
Output format
Subtasks: 1, 2, 3, ...
Summary of day 1: ...
Summary of day 2: ...
Summary of day 3: ...
Summary of day 4, 5 etc: ...
Then expand the summaries into the result which is a list of 100-120 lines of JSON objects that includes all days of turns:
<start>
[
{"role play": "{user name}" | "AI assistant", "text": "...", "ts": "YYYY-MM-DD HH:MM"}
... (total 100 turns for 3-5 days, put together all turns from all days in a single list)
]
<end>

*USER PROMPT*:

Last generation: "{last generation}"
Reflection: "{feedback}"

## E.2 Prompts for HomerAgents-Neo

> **Rules for Tasks Generation in HomerAgents-Neo**
>
> - The task description should be a string that describes each subtask (1-5 subtasks) to be completed.
> - Only follow and use the evaluation criteria formatted from the examples and do not invent new evaluation criteria.
> - The evaluation criteria should be a list of dictionaries, each dictionary representing an evaluation
> - The task description is hidden from the agent, and a ground truth agent should be able to complete the task with just the task description.
> - The ground truth memory should contain the necessary facts (things like time, new values, new filenames, new content values (but intermediate or final calculations), etc.) and events (action items) needed to complete the task, which will be distributed across the chat histories. These memories when disepensed across the chat histories, should be related to the task and queryable using the query sentence.
> - For the query sentence, it should be a general instruction of the task description, which will be sent to the policy agent to understand the general task and use it to query more details about the task details from memories.
> - FOR EXCEL TASKS, we do not have ground truth reference files, DO NOT USE evaluate exact match with a reference excel file. Instead, use the evaluation criteria to check some important added values to the excel such as {{"function": "evaluate excel cell value","args": {{"file": "data/salary.xlsx","matches": [{{"row": 5,"col": 2,"value": "200000"}}]}}}}, etc.
> - FOR CALENDAR TASKS, the commands for creating calendar events do not contain information such as one hour reminders or locations, so do not use these as task or evaluation criteria. Instead, if you want to evaluate these, use the event's title, start time and end time as evaluation. If you want to evaluate the event's details such as location, ask the agent to add these details to the event title, and add this action item note to the ground truth memory for chat generation. Note that when generating a task, you should be precise about what to expect for the calendar's description as an LLM policy agent may generate events with different names.
> - FOR QUESTION ANSWERING TASKS, expect the agent to output a the final answer in the answer.txt file, instead of adding a line in an existing file like word or excel file. When evaluating such answers, be precise about the task, ground truth memory such that you can expect what the agent produce so that the correctness of the answer is easily verifiable.
> - The inference agents can create or modify files such as docx, xlsx, generate pdf files. No powerpoint or txt files are allowed except for the answer.txt file where the policy agent's final output is logged. 11. FOR EMAIL TASKS, there is no draft mode or attachment options. Follow closely the examples given below, and do not create new evaluation criteria formats.
> - FOR WORD (docx file generation or update) TASKS such as summarization, evaluation on a subset of the most important keywords is sufficient and do not match the exact content or long sentences as the inference agent are not expected to generate the exact matches.
> - As a general rule, make sure that the facts and values, output file name and action items in the proposed task and memory are precise and clear and matches the evaluation criteria accurately, such that the agent can accurately complete the task. If you leave the task description vague, the agent may write to wrong file names, wrong event details, etc. For example, for setting up a calendar event, make sure you specify the exact start time and end time, and the exact description of the event, so that the agent can create the event with the correct details. For creating new files, make sure you specify the exact file name, etc. And make sure that these important points or action items are clearly described in the ground truth memory so that an inference agent with query sentence and ground truth memory can complete the task as in the task description.
> - Provide new and complementary information about your proposed new tasks in the ground truth memory, and DO NOT INCLUDE the solution to the task such as the intermediate steps for the solution (such as values read from files or intermediate or final calculated values), but rather a list of facts and action items that are necessary for completing the task complementing the files, such as missing details from the files, important action items or notes missing in the query sentence such as the output filenames, locations to put values, what elements a calendar event description should contain, or new events you propose or new facts. The memory generated will appear in the chat histories. The inference agent has access to all the files, and should be able to query the ground truth memory using the query sentence to find the necessary facts and action items to complete the task, while the query sentence should miss some details such as facts or preferences, which can be found in the memory.
> - Follow closely the json format and function names in the given examples when generating evaluations and do not invent new evaluation functions, and for keyword checks, split those keywords into different chunks to avoid being too strict (e.g., split and skip the punctuation marks).

---

**Rules for Dialogues Generation in HOMERAGENTS-NEO**

- The generated chat histories should contain around 100-120 turns per day, spread across 5 days (before today).
- To generate the chats, Take the following steps as the orchestrator: #### Break Down Memory per Chat Day: First extract the precise subtask action items or the factual knowledge from the ground truth memory pieces to be covered for each day. #### Chat Generation: For each day (day 0 to day 4), provide the PRECISE memory pieces for the day as the orchestrator, and ask the chat generator agent to write the chat histories day by day using the ChatTool. For example: to generate day N chat history with chat generator agent, first extract and mention the list of ¡EXACT MEMORY CONTENTS¿ to be covered on the day and let it generate chats that precisely capture these contents. Make sure that with the memory pieces, the inference agent can find the action items to work on, the correct file names, and the correct content values to complete the task. Beware that sometimes if the description is vague, the agent may write to wrong file names, wrong event details, etc. #### To make the chat histories longer, chitchat with the agent that are not related to the task can be added, but make sure that these do not add noise to the task solving such as new action items that are not covered by the memory or task description. Do not duplicate the memory pieces across the chat days, and if all memories have been covered, the chat history of the next day can be just about chitchat.
- Each chat turn being a json object with timestamp, the source (user or agent), and the content.
- The chat is between the user and the agent (not human), the user may mention the facts from the memory or action items from the task description, and the agent may respond with answers like will do but not solve the action, so that during inference, the agent can find the action items to work on.

### E.2.1 TASK GENERATOR PROMPT

---

**Prompt 3: Task Generator Prompt**

*SYS PROMPT*:

Generate a task description, evaluation criteria, and ground truth memory for the task. Use the TaskTool to log it. The task description should be a string, the evaluation criteria should be a list of dictionaries, each dictionary representing an evaluation criterion, and the ground truth memory should be a list of dictionaries, each dictionary representing a memory item. Use double quotes and not single quotes. The format of the arguments to the tool call to the tool named TaskTool should be: {'task_specs': <the json object with task description, evaluation criteria, query sentence, and ground truth memory'>} where the tool name is TaskTool. NOTE THAT the json object should be valid with double quotes on the keys and values.
**{Rules for Tasks generation}**

---

*USER PROMPT*:

Context: "{context information}"
Instruction: "{instruction from orchestrator}"

---

### E.2.2  ORCHESTRATOR PROMPT

---

**Prompt 4: Orchestrator Prompt**

*SYS PROMPT*:

Today is {date} ({weekday}). The current time is {time}. You are an AI assistant for user {username}. Now you're the orchestrator and your task is to synthesize new tasks to evaluate agents' memory capability for task solving. To generate this new task, you will need to generate task specifications and chat histories which includes important memory of information for solving the task, please follow the following steps:

###STEP 1: FILE READING: First start by reading some existing files (such as excel, email, calendar, or other files) using the file related task agents, it is possible there are sometimes no files while it is still possible to propose tasks. Gather important information from these files that are related to the task you want to propose, such as where to update a file or to use information from these files. Note that the inference agent will have access to these files, so the ground truth memory and the chat histories to generate is not just about recording specific elements in the files but more about new information or action items relates but not limited to contents already in the file.

###STEP 2: TASK PROPOSAL: then propose a new task which includes information:
1. a task description (hidden from agent),
2. the task evaluation criteria (hidden from agent),
3. a ground truth memory which includes facts and events needed to complete the task (hidden from agent), 4. a query sentence which is a more general instruction of the task description which will be sent to the policy agent to understand the general task and use it to query more details about the task details from memories. 5. for evaluation, txt is not a file format that can be used, so please do not generate tasks that require generating new txt files. For safe evaluation, please follow the task spec examples below to generate possible tasks and evaluations.{task_spec_examples}

###STEP 3: LOG DOWN THE TASK SPECS: After proposing the task, use the task_generator_agent to write down these task details using the TaskTool.

###STEP 4: GENERATE DIALOGUES: (DO NOT FORGET) After generating the task, expand the ground truth memory into long chat histories where the ground truth memories are scattered, such that during inference, the agent can be challenged on curating correct pieces of memories from these chats.

**{Rules for Tasks generation}**
**{Rules for Dialogues generation}**

As a general note, you can find files, calendar events, emails for your task in '/testbed/data', you can use the assistant agents to read, list, the files, do not create new items for this task generation cycle.

DO NOT TERMINATE THE TASK IF YOU HAVE NOT FINISHED GENERATING THE TASK SPECS OR THE DIALOGUES. DO NOT STOP TO GET HUMAN FEEDBACK, JUST GENERATE THE TASK SPECS AND DIALOGUES.

---

*USER PROMPT*:

Context: "{context information}"

### E.2.3   CHAT GENERATOR PROMPT

---

**Prompt 5: Dialogue Generator Prompt**

*SYS PROMPT*:

Today is {date} ({weekday}). The current time is {time}. You are an AI assistant for user username. Now you're the chat generator assistant helping a task generator orchestrator to synthesize new tasks. Your job is to expand the ground truth memory into chat histories where the memories are scattered in the chat histories. Generate chat histories for the task given the ground truth memory and task description.
**{Rules for Dialogues generation}**

---

*USER PROMPT*:

Task: "{task}"
Subtask Instruction: "{subtask instruction}"
Instruction: "{instruction from orchestrator}"

