# OpenReview forum: "OdysseyBench: Evaluating LLM Agents on Long-Horizon Complex Office Application Workflows"
_ICLR.cc/2026/Conference — ICLR 2026 Conference Withdrawn Submission_

### Official Review · Reviewer_ajcR · 2025-10-19

**Soundness:** 2
**Presentation:** 2
**Contribution:** 2
**Rating:** 2
**Confidence:** 3

**Summary:**

This paper introduces OdysseyBench, a comprehensive benchmark designed to evaluate large language model (LLM) agents on long-horizon, multi-application workflows in realistic office scenarios such as Word, Excel, PDF, Email, and Calendar. It consists of two complementary splits—OdysseyBench+ (300 real-world tasks) and OdysseyBench-Neo (302 synthesized complex tasks). To enable scalable benchmark construction, the authors propose HOMERAGENTS, a multi-agent framework that automates environment exploration, task generation, and dialogue synthesis.

**Strengths:**

1. This paper proposes a new benchmark, OdysseyBench, to evaluate the comprehensive capabilities of LLM agents in long-term office tasks, and covers a variety of real-world office applications: Word, Excel, PDF, Email, and Calendar.
2. The authors also propose HOMERAGENTS, a multi-agent automatic generation framework for scalably building long-term task benchmarks. Through environment exploration, the task generation and dialogue synthesis stages are automated.
3. Long-Horizon Complex tasks assessment has some prospects.

**Weaknesses:**

1. The HOMERAGENTS framework is not particularly novel — similar approaches that combine information collection and data generation have already been employed in several prior works, such as AppAgent and others.
2. Long-horizon complex tasks are indeed underexplored in previous datasets and benchmarks. However, as a benchmark, it should be more comprehensive; the current limited amount of data is insufficient to support a thorough evaluation of agents’ capabilities in complex task scenarios.
3. The connection between HOMERAGENTS+ and HOMERAGENTS-NEO is not clearly explained, making the overall narrative behind the construction of these two splits somewhat confusing.
4. The experimental design is relatively weak, as many appropriate baselines were not included in the evaluation. Despite experimental analysis, no interesting conclusions were drawn.

More weaknesses can be found in the questions section for reference.

**Questions:**

1. Figure 1 presents a comparison; however, if the focus is on the shortcomings of complex tasks, I suggest differentiating between complex and long-horizon (memory-dependent) tasks within these benchmarks, rather than contrasting them with simple tasks.
2. OdysseyBench+ and OdysseyBench-Neo together form the entire dataset, but readers may wonder why the authors chose to distinguish between these two splits. My understanding is that they were constructed using different methods; however, the paper does not clearly explain why two separate data construction approaches were necessary.
3. Section 3.1 has already explained the overall process clearly, and the additional Algorithm 1 and Algorithm 2 do not help clarify the information. Instead, they make the paper look more like a method-oriented work. Overall, the writing of this paper has considerable room for improvement.
4. It may be necessary to consider more recent works. Given the applications chosen by the authors, this benchmark is more closely related to web-based environments. Therefore, at a minimum, the evaluation should include three types of web agents: web agent frameworks, RL-based agents, and the thinking models already covered in the paper.
5. For baselines, for example, [1] WebAgent-R1 and [2] WebPilot: A Versatile and Autonomous Multi-Agent System for Web Task Execution with Strategic Exploration should be considered as representative web-agent frameworks.
As for thinking models, it would be valuable to include recent ones such as doubao-1.5-thinking-pro, Claude-3.7-Sonnet-20250219, and Computer-Use-Preview for a more comprehensive comparison.
6. line 227: The writing does not conform to standard algorithmic conventions.
7. My understanding is that long-horizon tasks are closer to a combination of long-memory and informative search. Is that correct?

---

### Official Review · Reviewer_3KEf · 2025-10-26

**Soundness:** 3
**Presentation:** 2
**Contribution:** 3
**Rating:** 4
**Confidence:** 3

**Summary:**

This paper proposes a new benchmark (OdysseyBench) for evaluation of LLM-driven agents on complex, long-horizon workflows that involve multiple office applications. In addition, it proposes a multi-agent framework named HomerAgents for construction of benchmarks generally.

**Strengths:**

* The benchmarks contain both the manually-curated and synthesized queries, which is good for the evaluation of LLMs nowadays.
* The paper is generally well-written and easy to follow.

**Weaknesses:**

* The benchmark is built upon/inspired from the existing OfficeBench, but there exists no detailed analysis on: 1) the differences between the previous work and the current study; 2) the limitations of the previous benchmark and how they are handled in the current study; 3) the environment (infra) scaling issue long horizon, complex office task evaluation.
* There exists no discussions on the performance/accuracy of the generator and verifier itself for synthesis/extension of office tasks, similar for the homeagents-neo pipeline (planning/generation).
* The evaluation metrics are limited where mostly the pass rate/success rate is reported and such coarse-grained metric cannot indicate the future improvement for LLMs.

**Questions:**

* Some references are not valid (e.g., line 148-149).
* How to interpret Table 1? Is the minimal difference between 1-apps and 3-apps in human performance implying that the collected tasks are of low discriminability?
* What is the relevance between the proposed benchmark and existing ones? In other word, it is interesting to see if the conclusions from the proposed benchmark are consistent/contrasted with previous ones (e.g., officebench).
* The evaluation (long-context and rag-evaluation) is missing details. For examples, what is the effect embedding/summarization models (performance themselves) on the entire benchmark? What is summarization technique/prompt?
* If any agent-framework is involved in the evaluation?
* What is the representation of environmental feedback and how this might get improved for performance of agents?

---

### Official Review · Reviewer_vquL · 2025-10-27

**Soundness:** 2
**Presentation:** 2
**Contribution:** 2
**Rating:** 2
**Confidence:** 4

**Summary:**

This paper introduces OdysseyBench, a new benchmark designed to evaluate the capabilities of LLM agents on complex, long-horizon tasks within office application environments (Word, Excel, PDF, Email, Calendar). The authors argue that existing benchmarks (e.g., OfficeBench, AgentBench) primarily focus on "atomic", self-contained tasks, failing to test an agent's ability to maintain context, reason over long interaction histories, and handle dependencies across multiple steps.

The paper presents two main contributions:
1.  OdysseyBench: A benchmark with two splits: OdysseyBench+ (300 tasks) which transforms atomic tasks from the existing OfficeBench into long-horizon workflows by synthetically generating dialogue histories, and OdysseyBench-Neo (302 tasks) which are entirely new, complex tasks generated from scratch.
2.  HomerAgents: A multi-agent framework created to automate the generation of these long-horizon benchmarks. HomerAgents+ uses a generator-verifier setup for refinement, while HomerAgents-Neo uses a multi-phase (Planning, Exploration, Task Generation, Dialogue Generation) system to create new tasks.

The authors evaluate several proprietary and open-weight models on OdysseyBench, using both full long-context and various RAG settings. Their results show that these long-horizon tasks are significantly more challenging for current agents than atomic tasks, and they analyze the effectiveness of different context retrieval strategies.

**Strengths:**

1.  Addresses a Good question: The paper identifies a good limitation of current agent benchmarks: their focus on "atomic" tasks. It rightly argues for the need to evaluate agents on long-horizon workflows that require reasoning over accumulated
2.  Good Evaluation: The paper provides an evaluation of numerous models. The inclusion of both long-context and various RAG strategies (raw vs. summary, session-level vs. chunk-level) provides a valuable analysis of how agents cope with long context.
4.  Human-in-the-Loop Validation: The use of human annotators to verify task solvability (Table 3) is a strong point, ensuring that the OdysseyBench-Neo tasks are coherent and achievable.

**Weaknesses:**

1.  Overclaiming and Imprecise Language: The paper's repeated claim that `OdysseyBench+` is derived from "real-world use cases" is an overstatement. It is a synthetic transformation of another benchmark (OfficeBench). This linguistic imprecision ("real-world" vs. "realistic simulation") undermines the paper's claims about the benchmark's grounding.
2.  Misplaced Focus on Method over Benchmark: This is a benchmark paper, but the majority of the methodology (Section 3) is devoted to the HomerAgents generation framework. The analysis of the OdysseyBench benchmark itself (Section 3.2.2) is brief. The paper would be much stronger if it relegated the complex details of HomerAgents to an appendix and used the main body to provide a much deeper analysis of the benchmark's properties: e.g., a taxonomy of the long-horizon reasoning types required (temporal, causal, informational dependency), a deeper dive into the failure modes (Section 6 is a good start but brief), and a qualitative comparison of the task structures in OdysseyBench+ vs. OdysseyBench-Neo.
3.  Experimental Insights are Not Novel: The main conclusions from the experiments (Section 5) are not surprising: multi-app tasks are harder, full context beats RAG, and summarization helps RAG. These are all known findings. While they validate the benchmark, they don't offer new insights into agent behavior that one might hope for from a new benchmark paper.

**Questions:**

Please solve the weakness above

---

### Official Review · Reviewer_CRVd · 2025-11-01

**Soundness:** 3
**Presentation:** 3
**Contribution:** 2
**Rating:** 4
**Confidence:** 3

**Summary:**

This paper introduces OdysseyBench, a comprehensive benchmark for evaluating LLM agents on long-horizon workflows across 5 office applications (Word, Excel, PDF, Email, Calendar). Compared with previous works, which overreliance on self-contained atomic tasks, this paper proposes HOMERAGENTS, a multi-agent framework that automates the generation of long-horizon workflow benchmarks. OdysseyBench includes 300 real-use-case-derived tasks (OdysseyBench+) and 302 newly synthesized complex tasks (OdysseyBench-Neo). Experimental results show humans achieve ~91.5% accuracy on OdysseyBench, while top LLMs (e.g., o3, GPT-5) only reach ~55%, with agent performance declining as tasks involve more applications.

**Strengths:**

1. The benchmark construction approach is comprehensive: it not only derives tasks from existing task intents (via HOMERAGENTS+ transforming atomic tasks from OfficeBench into context-rich scenarios) but also generates long-horizon tasks entirely from scratch (via HOMERAGENTS-NEO). Notably, both subsets (OdysseyBench+ and OdysseyBench-Neo) are accompanied by multi-day dialogue histories, which better simulate real-world user-agent interactions.
2. Testing human performance is well done. Human annotators achieved an overall accuracy of 91.5% on OdysseyBench, which effectively validates that the benchmark is well-formed, solvable, and accurate.
3. Experiments on context management within OdysseyBench provide valuable insights for agent design. For instance, findings on the superiority of summarized context (over raw dialogue data) in enhancing reasoning efficiency, and the performance decline of agents when tasks involve more applications, offer valuable guidance for designing agents.

**Weaknesses:**

1. Ablation studies for the OdysseyBench-Neo construction process are lacking, and key concepts (e.g., "surfers") lack concrete case illustrations. The paper proposes HOMERAGENTS-NEO to generate long-horizon tasks from scratch via four phases , but it does not validate the necessity of each component or compare its performance against simpler generation methods. Additionally, the role of "surfers" in collecting contextual information from office applications is described in abstract terms, and it would be better to provide some specific cases.
2. The case study is underdeveloped. While the study identifies four agent failure modes (missing files, incorrect tool calls, etc.), it does not analyze whether these issues can be addressed by optimizing agent frameworks or by injecting office workflow knowledge into LLMs. Without comparing these potential solutions, the paper misses an opportunity to guide future research on bridging the performance gap between humans and LLMs on long-horizon office tasks.

**Questions:**

See weaknesses.
1. Ablation studies for the OdysseyBench-Neo construction process are lacking.
2. The case study is underdeveloped. More insightful guidelines are needed to steer future research toward bridging the performance gap between humans and LLMs on long-horizon office tasks.

---

### Note · Authors · 2025-12-11

I have read and agree with the venue's withdrawal policy on behalf of myself and my co-authors.